# Characterization of Ternary CuNiCo Metallic Nanoparticles Produced by Hydrogen Reduction

**DOI:** 10.3390/ma14206006

**Published:** 2021-10-12

**Authors:** Eliana Paola Marín Castaño, José Brant de Campos, Ivan Guillermo Solórzano-Naranjo, Eduardo de Albuquerque Brocchi

**Affiliations:** 1Department of Chemical Engineering and Materials Science, Pontifical Catholic University of Rio de Janeiro, Rio de Janeiro 22453-900, Brazil; guilsol@puc-rio.br (I.G.S.-N.); ebrocchi@puc-rio.br (E.d.A.B.); 2Department of Mechanical Engineering, Rio de Janeiro State University, Rio de Janeiro 20559-900, Brazil; brant@uerj.br

**Keywords:** Cu–Ni–Co alloy, characterization, hydrogen reduction, nanoparticles

## Abstract

Different methods of producing nanostructured materials at the laboratory scale have been studied using a variety of physical and chemical techniques, though the challenge here is the homogeneous distribution of the elements which also depends on the precursor elements. This work thus focused on the micro-analytical characterization of Cu–Ni–Co metallic nanoparticles produced by an alternative chemical route aiming to produce solid solution nanoparticles. This method was based on two steps: the co-formation of oxides by nitrates’ decomposition followed by their hydrogen reduction. Based on the initial composition of precursor nitrates, three homogeneous ternaries of the Ni, Cu and Co final alloy products were pre-established. Thus, the compositions in %wt of the synthesized alloy particles studied in this work are 24Cu–64Ni–12Co, 12Cu–64Ni–24Co and 10Cu–80Ni–10Co. Both precursor oxides and metallic powders were characterized by means of X-ray powder diffraction (XRD), scanning electron microscopy (SEM/EDS) and transmission electron microscopy (TEM). The results show that the synthesis procedure was successful since it produced a homogeneous material distributed in different particle sizes depending on the temperature applied in the reducing process. The final composition of the metallic product was consistent with what was theoretically expected. Resulting from reduction at the lower temperature of 300 ${}^{{}^{\circ}}$C, the main powder product consisted of particles with a spheroidal and eventually facetted morphology of 50 nm on average, which shared the same FCC crystal structure. Particles smaller than 100 nm in the Cu–Ni–Co alloy agglomerates were also observed. At a higher reduction temperature, the ternary powder developed robust particles of 1 micron in size, which are, in fact, the result of the coarsening of several nanoparticles.

## 1. Introduction

Alloy development has been imperative in our modern industrial scene: from infrastructure to informatics, as well as from energy and environment to medicine and aerospace [1,2]. For instance, nickel, cobalt, and copper are metals with good catalytic, electronic, and magnetic properties [3], and when these metals are mixed and transformed into binary or ternary alloys, novel and more powerful properties may be attained.

It is also claimed that a variety of Cu–Ni–Co alloys can be successfully used in several applications since the amount of each component interferes in the alloy properties [4]. For example, small amounts of cobalt integrated into copper, nickel, or copper–nickel alloys provide magnetic or GMR materials [5,6,7,8]. Additionally, a high degree of hardness can be achieved if the alloy contains fine grains [9,10], and when the cobalt is present in abundance, corrosion resistance can be deteriorated [11,12].

Moreover, it is well established that alloys may enormously increase their effectiveness when nanostructured [5,13] and in recent years, the study of Cu–Ni–Co ternary alloys at the nanoscale has increased, as there is great interest in new potential applications [4,5,14,15,16]. An example is the current interest in materials with magnetic properties and data storage capacity for application in hard disk drives and sensors [17]. This specific alloy’s giant magnetoresistance (GMR) property has been widely studied [5,14,15]. Although at a smaller scale, the Co–Ni–Cu alloy nanocrystals’ ability as a catalyst has also been investigated [17].

However, it is well known that one of the challenges of obtaining nanoparticles’ alloy content is the production process and thus, different methodologies have been investigated, which can be categorized as either chemical or physical based techniques. There is electrodeposition [5,6,18,19,20], vapor deposition [21,22], mechanical alloying [3], synthesis by hydrazine reduction [17], positive microemulsion [23], and melt spinning [15]. Electrodeposition is the most common studied method which has allowed to obtain uniform, smooth and fine-grained alloys with adequate morphology for specific applications [18]. Nonetheless, it incorporates a certain complexity that complicates its application at a large scale [5,14,16,17,18,20]. The hydrogen reduction of homogeneous co-precipitated oxides has been mentioned as an alternative method to obtain binary alloys [24,25,26] and nanocomposites [27,28,29,30], but it has not yet been studied as a method of obtaining ternary alloys. The synthesis process stands up as a simple two-step method: (1) the generation of a co-formed homogeneous oxides mixture from a decomposition of a pre-established nitrates solution followed by (2) the hydrogen reduction of the co-formed oxides at relatively low temperatures. By applying this methodology, it is possible to obtain ternary alloys with different compositions.

In this context, the present work aimed to characterize the precursor (co-formed oxides) and the Cu–Ni–Co alloys produced by the aforementioned two-step method in order to verify whether this can provide the required nanoscale particles as well as the desired homogeneous distribution.

## 2. Materials and Experimental Procedure

The precursors were copper nitrate hexahydrate Cu(NO${}_{3}$)${}_{2}\cdot 6$H${}_{2}$O (99%), nickel nitrate hexahydrate Ni(NO${}_{3}$)${}_{2}\cdot 6$H${}_{2}$O (99%) and cobalt nitrate hexahydrate Co(NO${}_{3}$)${}_{2}\cdot 6$H${}_{2}$O (99%), all from Sigma-Aldrich^®^ (Burlington, MA, USA). The experimental procedure was carried out by dissolving the nitrates in a homogeneous aqueous solution, in previous established compositions, followed by a complete thermal decomposition. Then, the obtained material, a multi-oxide compound, was reduced by hydrogen. Both steps were always confirmed by weight loss in comparison with the expected stoichiometric for the respective reactions (oxides’ formation and reduction). The obtained metallic powder samples were then cooled down over the same controlled hydrogen atmosphere until they achieved room temperature in order to avoid the re-oxidation of the sample.

The favorable thermodynamics conditions for these two steps (thermal decomposition and hydrogen reduction) can be seen in Figure 1a,b, which show the existence of negative values for $\Delta G^{{}^{\circ}}$ within a large temperature range. The thermal decomposition temperature was established at 500 ${}^{{}^{\circ}}$C due to kinetics reasons while the reduction temperature was tested at 300, 600 and 900 ${}^{{}^{\circ}}$C.

Interestingly, exhaustive studies regarding this alternative method—including the thermodynamics and kinetics of hydrogen reduction for different systems—have been reported [24,25,26,27,28,29,31]. Nevertheless, a general schematic vision is illustrated in Figure 2.

### Characterization Methods

Oxide and powder metal samples were analyzed by XRD in order to determine the crystal species. The employed diffractometer was a Bruker D8-Discover with a Bragg–Brentano geometry (coupled θ–2θ) (Billerica, MA, USA), a copper tube (λ = 1.5418 Å) working at 40 kV and 40 mA, and a LynxEye detector (Billerica, MA, USA). The scan was performed from 10${}^{{}^{\circ}}$ to 90${}^{{}^{\circ}}$ during 1 s with 0.02${}^{{}^{\circ}}$ 2θ step. Quantitative Rietveld fundamental parameters (FPs) calculations [32] using TOPAS 5.0 (Billerica, MA, USA) were performed for phase determination. For these calculations, the lattice parameter, the crystalline size and the scale were adjusted where the value of the former parameter indicated the weight phase percentage. The best adjustment was defined by the goodness of fit (GOF) parameter.

In order to analyze the oxide and metallic powder structure of the surface and elemental analysis, SEM was systematically employed and performed with the JEOL-JSM 6510/LV instrument (Akishima, Tokyo, Japan) at 15 kV using secondary electron and backscattering electron imaging detectors attached to an Oxford silicon drift detector of 8080 mm${}^{2}$ for EDS analysis.

For the TEM analysis, a diluted suspension of powder samples was submitted to ultrasound and drop deposited on an amorphous holly-carbon-coated standard Cu grid. The particle size and its morphology were measured by TEM imaging. Energy dispersive X-ray spectroscopy (EDS) was also employed to detect local compositions. These analyses were conducted in a field emission JEOL JEM 2100F instrument (Akishima, Tokyo, Japan) operating at 200 kV in diffraction contrast mode.

## 3. Results

### 3.1. Characterization of the Powdered Oxides

The powdered oxides, characterized by XRD, MEV/EDS, were the precursors of each metal alloy and were assigned according to their composition as follows: 24Cu–oxide (for the 24Cu–64Ni–12Co alloy composition); 12Cu–oxide (for the 12Cu–64Ni–24Co alloy composition); and 10Cu–oxide (for the 10Cu–80Ni–10Co alloy composition).

The XRD measurements were performed on all three samples obtained by thermal decomposition at 500 ${}^{{}^{\circ}}$C for semi-quantitative phase evaluation and the Rietveld method was applied for phase quantification, as shown in Figure 3. The black curve indicates the experimental data and the red one indicates the calculated data. One can see a good signal-to-noise relation in the experimental data that improved the goodness of fit (GOF) of the Rietveld calculations, which varied between 1.1 and 1.3. As can be noted, this difference was very small, leading to a lower deviation in the phase quantification.

From XRD Rietveld calculations, it was possible to identify some of the content phases in the co-formed oxide samples. The main peaks and their intensities are highlighted by discontinuous lines. The identified phases NiO, Co${}_{3}$O${}_{4}$, Co(NiCo)O${}_{4}$, Ni${}_{80}$Cu${}_{2}$O, and NiCo${}_{2}$O${}_{4}$—together with their corresponding weight percentages—are indicated in the upper right side of the figure. Similar phases were detected in all analyzed samples. The higher intensity peaks shown in the XRD pattern were indexed as 022, 311, 222, 004, 022, 311 and 222. It was also noticed that other peaks such as 111, 422, 511 and 044, saw their intensities decreased, which was associated with the decreased cobalt quantity in each sample. It can be said that a complete nitrates decomposition was achieved since only oxide phases were detected. The average crystallite sizes from all three samples were below 50 nm.

As the SEM analysis of the co-formed oxides presented similar particle characteristics, Figure 4 illustrates, as an example, the image of one sample (10Cu–oxide). It shows small heterogeneous particles configuring porous aggregates. This characteristic is a desired and important advantage for solids to be submitted to hydrogen reduction as it is a gas–solid reaction. The elemental content of O, Ni, Cu, and Co were semi-quantified by EDS. Their estimated quantities are presented in weight percent and atomic percent in Table 1. The elementary weight percentages in all oxide samples were very similar to the expected theoretical values, a fact that once again corroborates the complete decomposition of nitrates.

Bright-field TEM imaging and the EDS elemental mapping of the same co-formed oxides sample is presented in Figure 5. Each element was associated with a color, with Ni as green, O as yellow, Co as violet, and Cu as red. All the elements display a uniform distribution, suggesting good homogeneity, which is important for the alloys’ formation by hydrogen reduction. In Figure 5b, four points mark the position of the electron beam where EDS data are acquired. The corresponding detected elemental values are shown in Table 2. Ni, Co and Cu, although not homogeneous as a whole, exhibited the expected amounts. However, the oxygen content shows higher values than expected for all marked points, probably due to the hydrophilic property of the oxide product and the conditions under which the analysis was conducted. The bright-field TEM image of a 10Cu–oxide sample, as an agglomerate of particles, is shown in Figure 6a. The central dark field image of Figure 6b, with the operated beam marked with the red circle in the diffraction pattern of Figure 6c, enables the isolation of the image of a single particle and the measurement of its size—in this case 66 nm. This particle size is comparable to the crystallite sizes determined by Rietveld calculations. An individual particle may thus have a content of one or two crystal domains.

### 3.2. Characterization of Ternary Alloy Metallic Powders

Metallic powders, after the reduction process, were characterized by XRD, SEM, TEM and EDS techniques. Using the same procedure applied to the oxide samples, the XRD measurements with the Rietveld method were performed on all three samples obtained by reduction at 900 ${}^{{}^{\circ}}$C (24Cu–64Ni–12Co, 12Cu–64Ni–24Co, 10Cu–80Ni–10Co) for qualitative phase evaluation, as shown in Figure 7.

From the diffraction peak indexing, one can see that 111, 200 and 220 peaks are the main characteristic peaks of the FCC phase. This result suggests that the three metal powders, under these compositions, may constitute a single FCC phase, with the same matrix as in the three alloys. The same XDS result for a Cu–Ni–Co alloy produced by different methodologies can be found in the literature, where it the presence of a similar FCC phase is mentioned [6,8,11,14,19,20]. The calculations of the Rietveld method were not accomplished due to the lack of reported phases that contain these elements (Cu, Ni and Co) in the Inorganic Crystal Structure Rietveld’s Database (ICSD). However, as an alternative option, the CIF files of CuNi, (Co${}_{3}$Ni${}_{7}$)${}_{0.4}$ and Ni were used in the Rietveld calculations. All of them are of FCC crystal structure. Through these results, it was possible to confirm the presence of these phases. The goodness of fit (GOF) of the Rietveld calculations was 1.1, which reveals the excellent quality of the phase identification calculations. Furthermore, from these calculations, it was possible to determine the average crystallite size between 118 nm and 140 nm. Numerous crystallite sizes ranging from 50 nm to 500 nm have been described in the literature, indicating that these sizes are process-dependent [4,8,17,18].

Figure 8 shows the reduction temperature effect on the morphology of the obtained metallic samples. It can be seen that the morphology of the metallic particles resembles those of the co-formed oxide particle powders since the same porous surfaces were observed in the metallic nanoparticles’ aggregates. Other types of morphologies have also been reported in the literature, such as “spherical-type” [5,17] and “large fiber-type” [4] morphologies. Here, the effect of the reduction temperature on the kinetics of particle growth and coarsening is clear. The surface morphology of the sample from a reduction at 600 ${}^{{}^{\circ}}$C (Figure 8b) reveals particles bonding with each other, developing necking as a self-sintering phenomenon in the absence of an externally applied pressure as well as particle coalescence. At the highest reduction temperature (900 ${}^{{}^{\circ}}$C), complete particle sintering and coalescence to micron sized particle aggregates can be observed (Figure 8c). An increase in the Cu content in the metal powder promotes the nanoparticles’ self-sintering and the further coalescence of the obtained ternary alloy powder—possibly due to its lower melting point (see Table 3). A similar phenomenon was observed by [20] as it is reported that the presence of copper may cause a nucleation effect during deposition.

The elemental mapping of the 10Cu80Ni10Co metallic powder reduced at 300${}^{{}^{\circ}}$C is shown in Figure 9. A good homogeneous distribution of the elements was achieved, although a Co-rich-segregated zone seems to appear in the central region of this particle (the nanopowder aggregate). The CDF image of Figure 9a, shows that this micron-sized particle is able to sustain crystal defects, namely dislocation, which are better resolved along the bend contour. In the CDF image of Figure 9b, the missals-like contrast reveals nanoparticles of a second phase that are two orders of magnitude smaller in size than the parent single particle—probably in the detected Co-rich phase.

Figure 10 is a TEM bright/dark-field image and diffraction pattern of the co-formed ternary powder (10Cu–80Ni–10Co). It is a composition of a bright field (BF/TEM) image of a robust single particle (reduction at 600 ${}^{{}^{\circ}}$C), probably resulting from smaller particles’ coalescence; two central dark field (CDF) images taken under slightly different incident electron beam/particle angles; as well as the corresponding electron diffraction patterns of the same particle. While the BF image mostly reveals the particle size, morphology and some diffraction contrast effects showing bend contours and some mass/thickness contrast effects, the CDF images bring additional information.

Figure 11 also shows a TEM bright field (BF)/centered dark field (CDF) image composition of a nanoparticles’ agglomerate resulting from the reduction process at 300 ${}^{{}^{\circ}}$C, with the selected area diffraction pattern, SADP, containing three nanoparticles. One CDF, shown in Figure 11c, allowed the imaging of a single nanoparticle, and thus its precise size measurement (27.13 nm). The other CDF image, shown in Figure 11d, was generated with two diffracted beams, circled in the SADP, and thus imaged two-facetted nanoparticles at the moment of necking.

The particles from the 300,600 ${}^{{}^{\circ}}$C reduction temperature, were, on average, smaller than 150 nm. The detailed diffraction contrast allowed observing the morphology and substructure of some individual co-formed ternary particles, and then, robust particles of 1 micron size were, in fact, the result of coarsening several nanoparticles. This fact is corroborated by its selected area diffraction pattern. Finally, the centered dark field TEM images have shown that individual nanoparticles tend to facet in order to minimize surface energy, prior to the process of self-sintering/coarsening into larger agglomerates, driven by surface energy minimization.

## 4. Conclusions

Ternary Cu–Ni–Co alloys were obtained by an alternative chemical procedure conducted by two steps: the thermal decomposition at 500 ${}^{{}^{\circ}}$C of a homogeneous nitrates solution in order to produce the co-formed oxides, followed by their hydrogen reduction at 300, 600 and 900 ${}^{{}^{\circ}}$C.

The materials produced in each step were characterized by DRX, SEM, TEM-EDS elementary mapping techniques.

From the XRD Rietveld calculations, it was possible to identify some of the content phases in the co-formed oxides samples such as NiO, CO${}_{3}$O${}_{4}$, Co(NiCo)O${}_{4}$ Ni${}_{8}$Cu${}_{2}$O, and NiCo${}_{2}$O${}_{4}$. The SEM analysis indicated that, in general, the precursors (co-formed oxides) showed a homogeneous distribution of the elements and a porous structure, which are recommended for the following hydrogen reduction. Those crystallite sizes, also determined by Rietveld calculations, are comparable with the ones obtained from individual particles by TEM. It is thus possible to assume that those individual particles are composed of one or two crystal domains.

The ternary alloys were obtained with compositions very close to what was theoretically expected: 10Cu–80Ni–10Co, 12Cu–64Ni–14Co and 24Cu–64Ni–12Co. The average crystallite sizes, calculated by the Rietveld method, from all three samples was below 50 nm. A good homogeneous distribution of the elements was achieved and self-sinterization was noticed in samples from reduction conducted at the highest temperature. This effect was even more pronounced in metallic nanoparticles with a higher copper content. A prominent FCC phase was identified as the main metallic powder constituent and nanoparticles with apparent spheroidal morphology and a homogeneous distribution were also found by TEM analysis.

## Figures and Tables

**Figure 1 materials-14-06006-f001:**
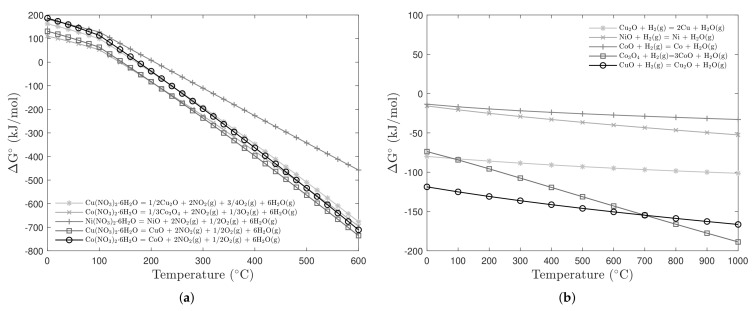
Thermodynamics diagram for (**a**) the thermal decomposition of nitrates; and (**b**) the hydrogen reduction of Cu, Ni and Co oxides.

**Figure 2 materials-14-06006-f002:**
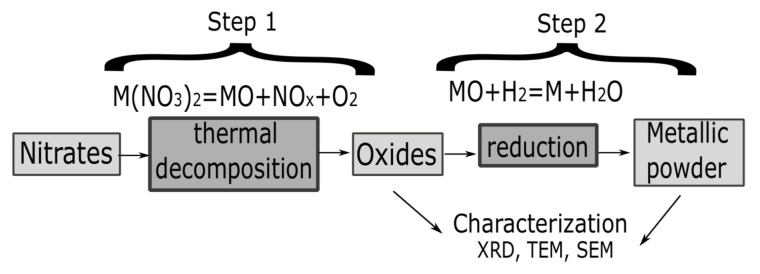
General schematic vision of the experimental procedure.

**Figure 3 materials-14-06006-f003:**
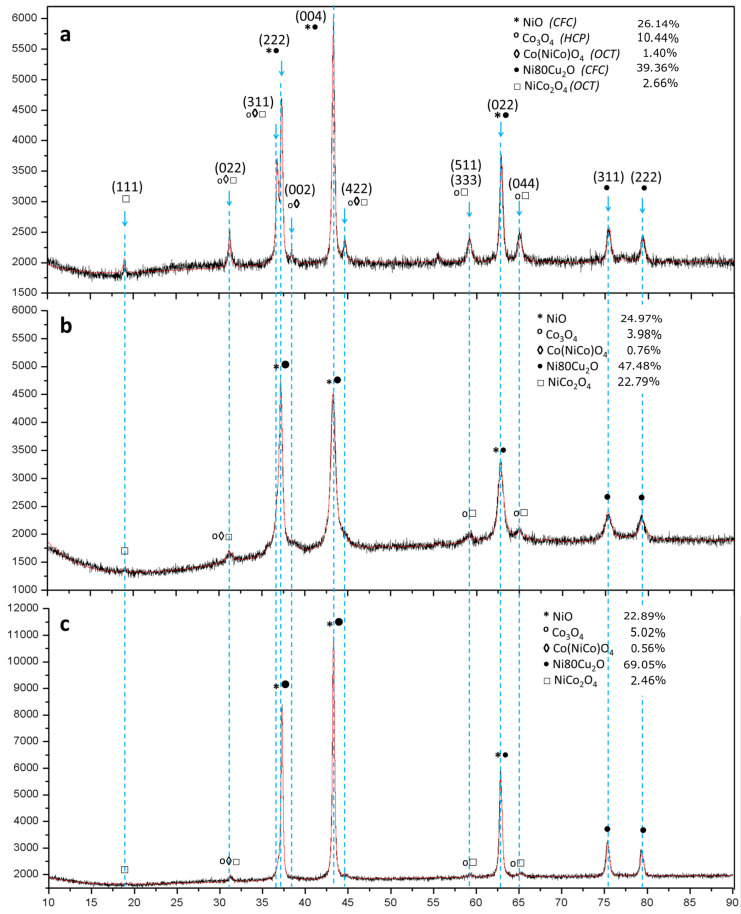
XRD of the co-formed oxides thermally decomposed at 500 ${}^{{}^{\circ}}$C: (**a**) 24Cu–oxide; (**b**) 12Cu–oxide; (**c**) 10Cu–oxide.

**Figure 4 materials-14-06006-f004:**
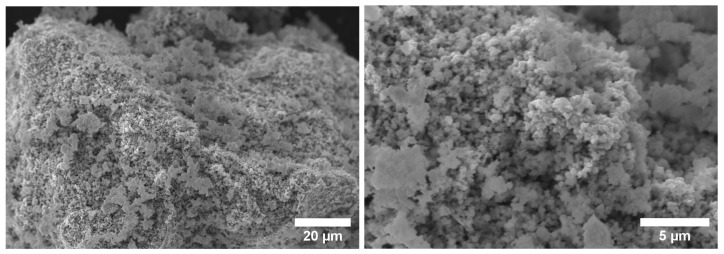
SEM analysis of the 10Cu–oxide sample thermally decomposed at 500 ${}^{{}^{\circ}}$C.

**Figure 5 materials-14-06006-f005:**
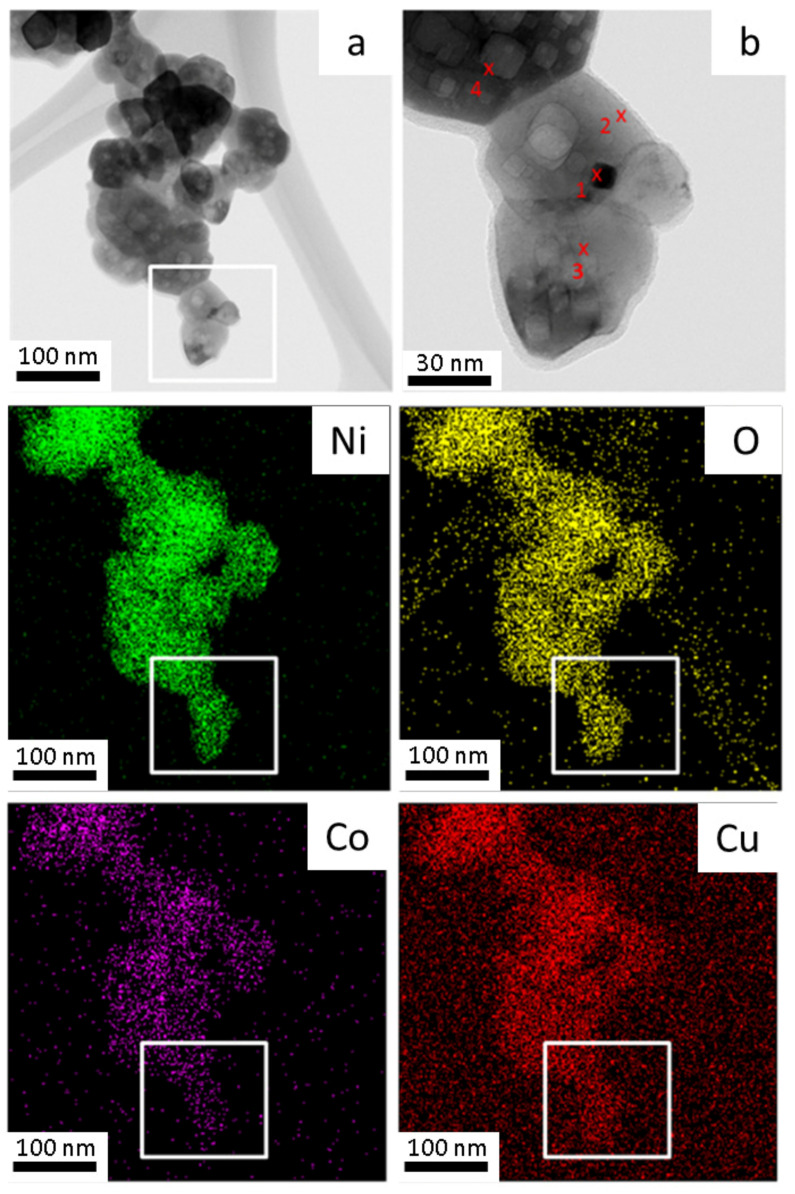
TEM bright-field image and elemental mapping of 10Cu–oxide nanoparticles (thermal decomposition at 500 ${}^{{}^{\circ}}$C) (**a**) metallic nanoparticles aggregate with the corresponding elemental EDS mapping shown at the bottom. The boxed area enlarged in (**b**) marks the positions of four-point analysis.

**Figure 6 materials-14-06006-f006:**
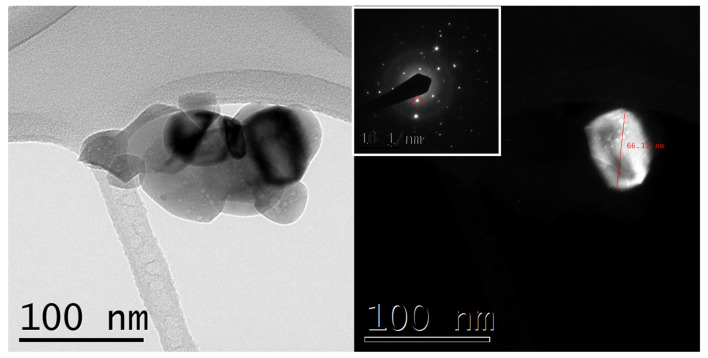
TEM bright-field image of a 10Cu–oxide nanoparticle aggregate and the dark field image of an isolated single crystal nanoparticle and the corresponding electro diffraction pattern.

**Figure 7 materials-14-06006-f007:**
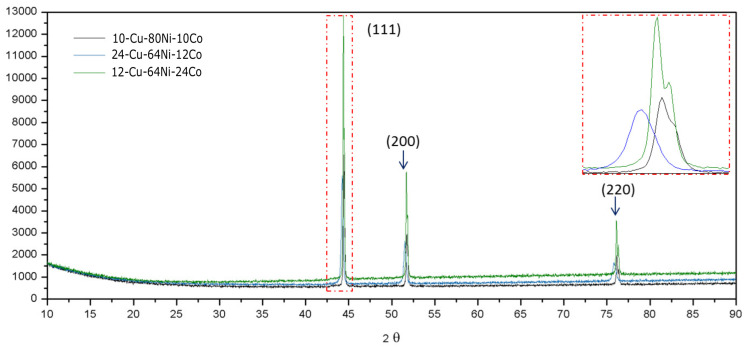
X-ray diffraction pattern of the ternary Cu–Ni–Co alloy powder in three different compositions (reduction at 900 ${}^{{}^{\circ}}$C).

**Figure 8 materials-14-06006-f008:**
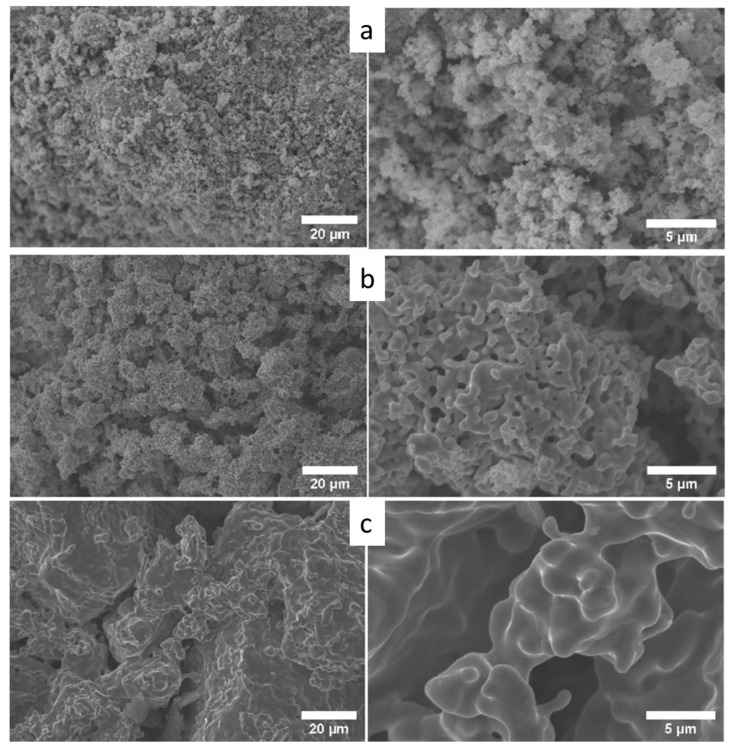
Secondary electrons SEM images under two magnifications of the co-formed ternary powder: 12Cu64Ni24Co for the different reduction temperatures: (**a**) 300 ${}^{{}^{\circ}}$C; (**b**) 600 ${}^{{}^{\circ}}$C; and (**c**) 900 ${}^{{}^{\circ}}$C.

**Figure 9 materials-14-06006-f009:**
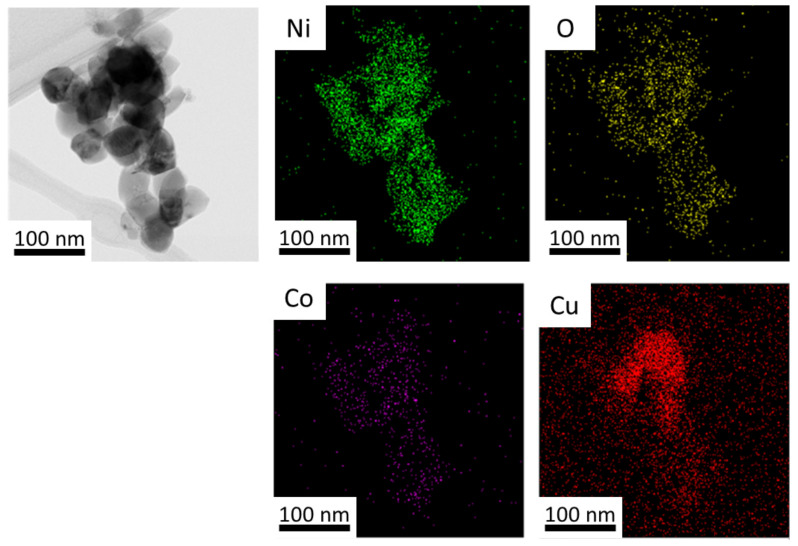
TEM bright-field image and elemental mapping of the co-formed ternary nanoparticle aggregate of metallic powder: 10Cu–80Ni–10Co (reduction at 300 ${}^{{}^{\circ}}$C/30 min).

**Figure 10 materials-14-06006-f010:**
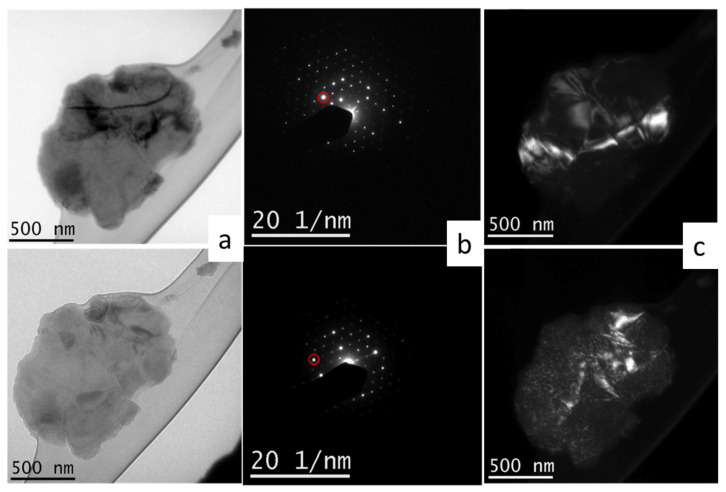
Two TEM bright/dark-field images of a co-formed and coalesced ternary nanoparticle aggregate: 10Cu–80Ni–10Co (reduction at 600 ${}^{{}^{\circ}}$C/10 min) obtained at a slightly different incident beam/sample orientation (**a**), the corresponding diffraction pattern (**b**) marking the operating reflection for the dark field images (**c**) revealing different portions of the nanocrystalline aggregate.

**Figure 11 materials-14-06006-f011:**
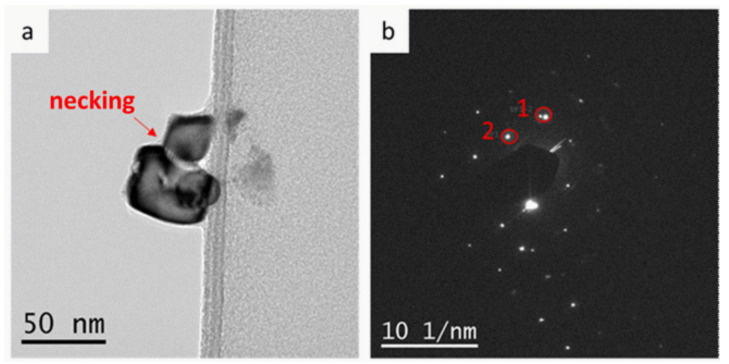
(**a**) TEM bright-field image of the co-formed ternary aggregate of three nanoparticles: 10Cu–80Ni–10Co and the corresponding selected area diffraction (**b**). The single reflection marked as 1 in this pattern generated the dark field image of the single nanoparticle of (**c**) and the double reflection marked as 2 generated the dark field mage of the two nanoparticles image of (**d**) (reduction at 300 ${}^{{}^{\circ}}$C/10 min).

**Table 1 materials-14-06006-t001:** EDS results for each measured point of the 10Cu–oxide sample of Figure 4.

O		Cu		Ni		Co	
**%wt**	**%at**	**%wt**	**%at**	**Ni %wt**	**%at**	**Co %wt**	**%at**
20.12	48.27	8.73	5.52	61.96	40.52	9.14	5.69

**Table 2 materials-14-06006-t002:** EDS results for each measured point of the 10Cu–oxide sample of Figure 5.

Point ↓	O		Cu		Ni		Co	
	%wt	%at	%wt	%at	Ni %wt	%at	Co %wt	%at
1	93.448	98.14	0.290	0.08	3.739	1.07	2.523	0.72
2	93.244	98.07	0.300	0.08	4.000	1.15	2.457	0.71
3	94.611	98.48	0.224	0.06	3.006	0.85	2.160	0.61
4	85.663	95.66	0.612	0.17	7.867	2.39	5.858	1.78

**Table 3 materials-14-06006-t003:** EDS results for each measured point of the 10Cu80Ni10Co sample.

Temperature ↓	O		Cu		Ni		Co	
	%wt	%at	%wt	%at	Ni %wt	%at	Co %wt	%at
300 ${}^{{}^{\circ}}$C	2.66	9.34	12.11	10.71	72.99	69.87	10.57	10.08
600 ${}^{{}^{\circ}}$C	1.26	4.51	10.75	9.69	76.11	74.26	11.88	11.54
900 ${}^{{}^{\circ}}$C	1.94	6.80	10.68	9.42	75.51	72.11	12.28	11.68

## Data Availability

The data presented in this study are available on request from the corresponding author.

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
