# Peer review of "Characterization of Ternary CuNiCo Metallic Nanoparticles Produced by Hydrogen Reduction"

_materials, 2021, doi:10.3390/ma14206006_

Round 1

Reviewer 1 Report

The authors report “Characterization of Ternary CuNiCo Metallic Nanoparticles Produced by Hydrogen Reduction”. Although the work presented is interesting to materials research community, this manuscript needs to be modified, please find the major concerns below.

  • Typically, SEM EDS is measured in atomic percentages. The authors measured/presenting in weight percentages. Why is that?
  • Can the authors estimate the crystallite sizes using Scherrer’s formula on XRD diffraction peaks? For example, see DOI: 10.1039/c8cc08101h and https://doi.org/10.1021/la0477183. Also compare the crystallite sizes with the TEM particle sizes to see it that matches, if not there might be polycrystalline particles present in the catalyst.
  • The authors mentioned in XRD results and discussion section that there are some diffraction peaks intensities are decreased. Did the authors observe any peak position shift compare to reference peak? A slight peak position shift is expected for the alloys depending on the size of the atom being incorporated into other crystal see for example, DOI: 10.1039/c8nr04399j.
  • STEM (bright filed TEM) images in Figure. 8 are plus to this manuscript, can the authors explain bit more in the discussion section? Can the authors show the image with all possible elements in it?
  • Does Figure. 11 really need to be under conclusions sections? Can this be moved up before conclusions?
  • Conclusions section needs to be re-written with no mention of any figure number and any general statements.

Author Response

Response to Referee #1

First of all, the authors are very grateful for the reviewer’s comments and critical assessment of the article. Therefore, all the points raised by them are answered bellow.

 Reviewer Point 1 - Typically, SEM EDS is measured in atomic percentages. The authors measured/presenting in weight percentages. Why is that?

Reply: We have prepared all the samples based on weight percentages. So, it was easier for us to compare the actual results with the predicted ones. However, the authors agree that SEM EDS is typically measured in atomic percentages. So, these results were also added in the revised manuscript (Tables 1 & 2).

Reviewer Point 2 - Can the authors estimate the crystallite sizes using Scherrer’s formula on XRD diffraction peaks? For example, see DOI: 10.1039/c8cc08101h and https://doi.org/10.1021/la0477183. Also compare the crystallite sizes with the TEM particle sizes to see it that matches, if not there might be polycrystalline particles present in the catalyst.

Reply: Yes! To estimate the crystallite sizes was a nice suggestion. Then, the use of the Rietveld Method has revealed that the crystallite sizes from Figure 3 - XRD results- are below 50 nm. Then, as suggested by the reviewer, it is comparable to the TEM particle sizes showed in Figure 5 and 6. Also, the crystallite size of the FCC phase, in Figure 7, ranges between 115nm and 140 nm. A number of comments related to this was added in different parts of the article.

Reviewer Point 3 - The authors mentioned in XRD results and discussion section that there are some diffraction peaks intensities are decreased. Did the authors observe any peak position shift compare to reference peak? A slight peak position shift is expected for the alloys depending on the size of the atom being incorporated into other crystal see for example, DOI: 10.1039/c8nr04399j.

Reply: The reviewer comment is correct and could be considered in the discussion. However, there is no notable increase and shifts of the diffracted peaks at the XRD results in Figure 7.

Reviewer Point 4 - STEM (bright filed TEM) images in Figure. 8 are plus to this manuscript, can the authors explain bit more in the discussion section? Can the authors show the image with all possible elements in it?

Reply: In the manuscript Figure 8 is related to SEM images and its presence aims to show the reduction temperature effect on the morphology of the obtained metallic samples, as discussed in the article. Perhaps the reviewer refers to Figure 9 (TEM bright field image) which already includes the compositional mapping requested.

Reviewer Point 5 - Does Figure. 11 really need to be under conclusions sections? Can this be moved up before conclusions?

Reply: It was a correct observation of the reviewer and has been adjusted in the revised manuscript.

Reviewer Point 6 - Conclusions section needs to be re-written with no mention of any figure number and any general statements.

Reply: The authors acknowledge the reviewer comment and the Conclusions section in the revised manuscript is not mentioning the figures.

Reviewer 2 Report

Manuscript ID:

Title: Characterization of Ternary CuNiCo Metallic Nanoparticles Produced by Hydrogen Reduction

Author(s): Eliana Paola Marín Castaño  , Ivan Guillermo Solórzano-Naranjo and Eduardo de Albuquerque Brocchi

The submitted manuscript reports on the synthesis and characterization of Cu-Ni-Co alloy metallic nanoparticles produced by the hydrogen reduction in two steps: i.e., firstly the co-formation of metal oxides by thermal decomposition at 500oC and then the hydrogen reduction of metal oxides at 300, 600 and 900 oC. The ternary alloy CuNiCo were characterized by scanning electron microscopy (SEM/EDS), transmission electron microscopy (TEM) and X-ray diffraction (XRD). Various morphologies were successfully obtained by applying different approaches. The synthesis of Cu-Ni-Co alloy nanoparticles showed a homogeneous distribution of elements, where the self-sintering was more effective with the high Cu concentration.

      Overall, the experimental results are well organized, however there still appear several issues in the submitted manuscript (refer to the comments below).  Especially, the discussion of abstract, introduction and result parts should be largely improved. The abstract is very general without the scientific discussion and not in flow. The introduction needs more description about the motivation behind this research work and should explain the main idea of paper. The RnD and characterization part should be discussed more properly with the scientific explanations and references. The article appears to be more like an experimental report without the proper scientific discussion. English and language should be reconsidered throughout the manuscript. Therefore, the reviewer considers the experimental work and discussion presented by the authors of insufficient quality to be suitable for the publication in the MDPI Materials in the current form and thus suggests a major revision.

Suggested decision: Major Revision.

Additional Comments:

  1. The abstract should be reconsidered. Instead of discussing characterization techniques, the main results should be discussed with a proper scientific description. The explanation is not up to the mark. There was no proper discussion about what is done in this work. The main purpose of an abstract is missing here. The description of the abstract should be an overview and the outcome of the summary work.
  2. The introduction part needs to reconsider, a logical flow is missing, and large improvement is needed with proper discussion and citations. The writing should be more scientific. It can explain why this research is important. The author talks a lot about the different methodologies to obtain nanoparticles content materials. It will be better to talk about the benefits of alloy CuNiCo nanoparticles. What are the applications? The scientific discussion for CuNiCo nanoparticles is required .The effect of alloy composition with the chemical property of the specific materials can be added to this part. Reconsider the English for the introduction part. Please discuss clearly the motivation behind this work and why this work has been done.
  3. It is hard to understand few figures without the proper labeling and descriptions. Data presentation can be improved.
  4. In the result section, XRD peaks should be mentioned with more proper discussion. XRD data should be discussed with proper scientific discussion and proper references should be cited.
  5. For the SEM analysis, the growth mechanism of nanoparticles should be properly discussed with a scientific explanation.
  6. Heat treatment effect on CuNiCo can be properly explained with a reasonable analysis.
  7. Generally, proper scientific explanations and citations should be provided with a logical flow.
  8. The author should reconsider the English language throughout the article. Please improve the overall text with the proper logic flow.

Author Response

Response to Referee # 2

First of all, the authors are very grateful for the reviewer’s comments and critical assessment of the article. Therefore, all the points raised by them are answered bellow.

Reviewer Point 1 - The abstract should be reconsidered. Instead of discussing characterization techniques, the main results should be discussed with a proper scientific description. The explanation is not up to the mark. There was no proper discussion about what is done in this work. The main purpose of an abstract is missing here. The description of the abstract should be an overview and the outcome of the summary work.

Reply: The authors acknowledge the reviewer comment and have re-written the Abstract accordingly to the suggestions made.

Reviewer Point 2 - The introduction part needs to reconsider, a logical flow is missing, and large improvement is needed with proper discussion and citations. The writing should be more scientific. It can explain why this research is important. The author talks a lot about the different methodologies to obtain nanoparticles content materials. It will be better to talk about the benefits of alloy CuNiCo nanoparticles. What are the applications? The scientific discussion for CuNiCo nanoparticles is required. The effect of alloy composition with the chemical property of the specific materials can be added to this part. Reconsider the English for the introduction part. Please discuss clearly the motivation behind this work and why this work has been done.

Reply: The authors acknowledge the reviewer comment and have re-written the Introduction, better explaining the advantages of the alloy system (properties and applications) and fields where the research can contribute for developing homogeneous nanostructured alloy products are mentioned. In that context, the CuNiCo ternary system was chosen and the characterization of nanoparticles produced by an alternative synthesis process is the main motivation of the article.  

Reviewer Point 3 - It is hard to understand few figures without the proper labeling and descriptions. Data presentation can be improved.

Reply: The authors acknowledge the reviewer comment and have improved the presentation, including new labeling (legends) for the figures.  

Reviewer Point 4 - In the result section, XRD peaks should be mentioned with more proper discussion. XRD data should be discussed with proper scientific discussion and proper references should be cited.

Reply: The authors acknowledge the reviewer comment and a more detailed discussion has been added in several parts of the revised manuscript related to the XRD results, including the use of the Rietveld Method to estimate the crystallite sizes.

Reviewer Point 5 - For the SEM analysis, the growth mechanism of nanoparticles should be properly discussed with a scientific explanation.

Reply: Again, the authors acknowledge the reviewer comment and a more detailed discussion has been added on Figures 10 and 11 of the revised manuscript.

Reviewer Point 6 - Heat treatment effect on CuNiCo can be properly explained with a reasonable analysis.

Reply: The temperature effect mentioned in the article refers to the hydrogen reduction process and its influence on the obtained material morphology. Actually, no heat treatment was applied on the metallic ternary product.

Reviewer Point 7 - Generally, proper scientific explanations and citations should be provided with a logical flow.

Reply: The authors agree with the reviewer comment and have tried to fulfill the suggestion while re-written the revised manuscript.

Reviewer Point 8 - The author should reconsider the English language throughout the article. Please improve the overall text with the proper logic flow.

Reply: The authors accepted the reviewer comment and have carried out a revision in the text of the revised manuscript.

Round 2

Reviewer 1 Report

The authors addressed the concerns raised in the previous revision and I recommend the manuscript to publish as is.

Reviewer 2 Report

The authors followed the reviewer's suggestions, and the paper can be accepted in the current form.